# Synergistic evolution of soil microaggregates biogeochemical processes driven by elevation gradients in Tongbai Mountain

Chunjie Li[1,2]*, Shili Guo[3], Songhao Shang[4]

1 School of Geographic Science and Tourism, Nanyang Normal University, Wolong Road No.1638, Nanyang, China, 2 Key Laboratory of Natural Disaster and Remote Sensing of Henan Province, Nanyang Normal University, Nanyang, China 3 School of Economics, Southwestern University of Finance and Economics, Liutai Avenue, Wenjiang, Chengdu, China, 4 State Key Laboratory of Hydro science and Engineering, Department of Hydraulic Engineering, Tsinghua University, Beijing, China,

* lichunjie08@126.com

## Abstract

The altitudinal gradient in mountainous areas triggers significant changes in landscape, climate, and vegetation, which in turn affect the vertical differentiation of soil type and their properties. Located in the transitional zone between the northern and southern climates of East Asia, Tongbai Mountain is of great geographical and ecological significance and is highly sensitive to global changes. However, systematic research on soil biogeochemical processes in the vertical zonation of mountainous areas in this transitional zone is still lacking. In this study, a sampling strategy based on altitude gradients was used, combined with a variety of advanced analytical techniques such as scanning electron microscopy (SEM), X-ray diffraction (XRD), X-ray fluorescence spectroscopy (XRF), and Fourier transform infrared spectroscopy (FTIR), to comprehensively characterize the mineral composition, soil organic carbon (SOC), and heavy metal element distribution of soils at different altitudes in Mount Tongbai. The results showed that the mineral composition of the soil exhibited a clear gradient based on altitude. The Quartz content was higher at low and high altitudes, while the feldspar mineral content was highest in the middle altitudes. Illite increased with increasing altitude. SOC content increased significantly with increasing altitude, and hydroxyl, amino and aliphatic organic matter were enriched in high altitudes. Heavy metal elements such as Fe, Ti, Cu, and Zn increased in high altitudes, and the microstructure of soil aggregates in high altitudes was more complex and stable. Through a comprehensive analysis, MgO, Zr, 2929 cm⁻¹, 3423 cm⁻¹, and Cu were selected as sensitive biogeochemical factors in the vertical band spectrum. This study reveals for the first time the mechanism behind the coordinated evolution of soil minerals, SOC, and heavy metal elements driven by the altitude gradient. This deepens our understanding of biogeochemical processes in the vertical band spectrum of mountains in the climatic transition zone between North and South East Asia, and

**Data availability statement:** All relevant data are included within the manuscript.

**Funding:** National Nature Science Foundation of China [grant number 41601614] The funders had no role in study design, data collection and analysis, decision to publish, or preparation of the manuscript

**Competing interests:** The authors have declared that no competing interests exist.

provides a scientific basis to formulate management strategies for mountain ecosystems in the climatic transition zone.

## 1. Introduction

Mountains are biodiversity hotspots and one of the most climate-sensitive ecosystems on the planet [1]. The altitude gradient in mountains leads to dramatic changes in landscape, temperature, precipitation, solar radiation, and vegetation [2], which in turn creates a vertical differentiation of soil types and properties [3]. The vertical zonation of mountains is an important feature of mountain ecosystems, reflecting the vertical distribution of environmental factors such as vegetation, climate, and soil with increasing altitudes. This vertical distribution not only shapes the unique ecological patterns of mountains, but also provides a natural experimental site to study the impact of environmental gradients on biogeochemical processes [4]. In mountain ecosystems, biogeochemical cycles play a vital role, regulating energy flow and material cycles, and influencing ecosystem structures and functions. With global climate change and the intensification of human activities affecting the environment, the biogeochemical processes of mountain soils and its feedback to the ecosystem have become topical in academia [5]. As a core component of Earth's critical zone, the structure and function of soil exhibit extremely complex characteristics from the nano to macro scale [6]. Soil aggregates, as the basic units of soil structure, play a key role in many ecosystem processes such as hydrological processes, soil erosion, microbial dynamics, and biogeochemical cycles [7–10]. Soil microaggregates are organic-mineral complexes formed through physical, chemical, and biological mechanisms, which in turn increase the stability of heavy metals in the soil [11]. At low temperatures (e.g., high-altitude soils), minerals are more stable in their association with soil organic matter (SOM) [12]. Most SOM is in contact with mineral surfaces, and this interaction plays a crucial role in the flow and transformation of carbon in the biosphere [13]. The interaction between minerals and SOM is not a static process, but a dynamic, multidimensional system. This signifies that minerals are not only SOM protectants but may also function as catalysts to promote SOM transformation and oxidation [7]. Organic-mineral complexes can significantly affect the migration and fixation of heavy metals, thereby affecting the chemical stability and ecological function of soil [14]. In recent years, important progress has been made in the study of soil biogeochemical processes and their driving mechanisms along elevation gradients. These new findings are of great significance for the understanding of complex processes in mountain ecosystems. Research in the Ecuadorian Amazon has revealed significant changes in soil physical properties with altitude, including dynamic changes in moisture content, particle structure, and mineral composition [15]. Mineral composition also undergoes a transformation with altitude, such as an enrichment of weathering-resistant minerals commonly found at high altitudes [16]. Soil organic carbon (SOC) generally increases with altitude because its decomposition rate decreases at higher altitudes due to lower temperatures, resulting in SOC accumulation [17]. The distribution of heavy metal elements in mountain soil

also shows a significant altitude gradient effect [2]. At high altitudes, due to lower temperatures and abundant precipitation, SOM enhances soil carbon storage capacity due to the adsorption and stabilization of clay minerals [18]. Related studies have shown that the altitude gradient has a significant impact on the structure and function of soil microbial communities and on the regulation of soil carbon and nitrogen cycles [4,19–21]. Soil is the largest terrestrial carbon pool on the planet, with a carbon storage five times greater than that of vegetation [19]. Soil carbon storage is one of the most important means of mitigating the increase in atmospheric carbon dioxide concentration caused by human activities [22]. Keller, Borer [22] emphasised how different soil textures and mineral structures affect soil carbon storage, particularly in the process of stabilising mineral-bound SOM.

As the source of the Huaihe River, the Tongbai Mountains are located in the transition zone between subtropical and warm temperate zones in East Asia, and are of great geographical and ecological significance [23]. The region is rich in biodiversity and species. It is also a key channel for gene exchanges between species in the north and south of East Asia. Due to its special geographical location and ecological characteristics, Mount Tongbai is highly sensitive to climate change. Small changes in the boundaries of climatic zones may cause significant changes in soil biogeochemical processes [24]. Although previous studies have explored the distribution of flora and the division of climatic zones in this region [25], systematic research on soil biogeochemical processes in the vertical belt spectrum of mountains in the north-south climatic transition zone is still lacking. Research on the biogeochemical characteristics of soil in the vertical belt spectrum of Mount Tongbai is of great significance to understand the soil nutrient cycle, element transformation, and environmental response in the north-south climatic transition zone. This study aims to fill this current research gap. This study is based on a sampling strategy with an altitude gradient and adopts a comprehensive multi-scale microscopic to macroscopic analysis method. This study comprehensively characterises soil samples using a combination of advanced analytical techniques such as scanning electron microscopy (SEM), Fourier transform infrared spectroscopy (FTIR), X-ray diffraction (XRD), and X-ray fluorescence spectroscopy (XRF) [26]. This study aimed to reveal changing patterns in the physicochemical properties of soil in the vertical zonal profile of the Tongbai Mountains, elucidating the distribution characteristics of heavy metal elements in soil and their relationship with the vertical zonal profile. Furthermore, it explored the mechanism of changes in SOC characteristics and functional group distribution with altitude, and comprehensively evaluated the impact of altitude gradients on biogeochemical processes in mountains. The innovation of this study lies in the combination of the vertical belt spectrum of the mountain with its biogeochemical processes, and a systematic study of the biogeochemical processes of mountain soils in the north-south climatic transition zone. Through multi-scale microscopic to macroscopic analysis, soil samples were comprehensively characterised, the interaction mechanism of soil minerals, elements, and SOC under the altitude gradient was revealed, and biogeochemically sensitive factors of the vertical belt spectrum were explored and selected. This study reveals the regulatory mechanisms of biogeochemical processes driven by altitude gradients, providing scientific evidence to establish a model of biogeochemical characteristics of soils in the Tongbai Mountain vertical zonation, and predicting the response of mountain ecosystems in the context of climate change. The results of the study can provide a scientific basis for mountain land management in the context of global climate change, the formulation of watershed management strategies in the Huaihe River basin, and the sustainable use and management of mountain ecosystems, resources, and the environment in the climatic transition zone between the north and the south.

## 2. Materials and methods

### 2.1. Overview of the study area and collection and processing of soil samples

Tongbai Mountain is located in the transition zone from subtropical to warm temperate climate in East Asia. The annual average temperature of Tongbai Mountain is 16°C, and the annual average precipitation is 900–1200 mm. It is located in the transition zone between the northern and southern climates of East Asia and is geographically significant [24]. Its geographical coordinates are roughly between 112°57′ and 113°30′ east longitude and 32°11′ and 32°29′ north latitude,

with an altitude of between 200 and 1,140 metres above sea level (Fig 1). Tongbai Mountain is the source of the Huaihe River, a major river in East Asia, and plays a prominent role in water conservation. Tongbai Mountain is rich in biodiversity and is the distribution boundary for many species. It is also an important channel for gene exchanges between species in the north and south of East Asia. The ecosystem in this transition zone is highly sensitive to global climate changes, and minor changes in the boundaries of climate zones may lead to significant changes in the ecosystem. Distributed above and below an altitude of 600 metres, the main vegetation type above 600 metres on Mount Tongbai is evergreen broad-leaved forest, while mixed coniferous and broad-leaved forest is the main type above 600 metres [24]. The parent material of soil is mainly metamorphic rocks and granites, and there is a lot of exposed mountain rock, which makes the soil relatively loose and soft. The soil pH value ranges from 4.8 to 6.8, which is acidic.

Field sampling was carried out with the approval of the Tongbai Mountain National Nature Reserve Administration Bureau, using an altitude gradient-based sampling strategy. Six sampling belts (A–F) were set up between 200 and 1140 m above sea level (Table 1), with six parallel sampling points in each belt. Surface soil samples were collected from a depth of 0–10 cm at each point. Five samples were collected from each point, mixed together, and then 500 g of the mixture was randomly selected for the experiment. The sampling was carried out in May 2023. The samples retrieved from the field were first frozen at −80°C for 1 hour, then subjected to low-temperature vacuum drying for 24 hours. Next, coarse particles and impurities were removed using a 2 mm sieve, and dried samples were ground to a fine powder. Finally, the

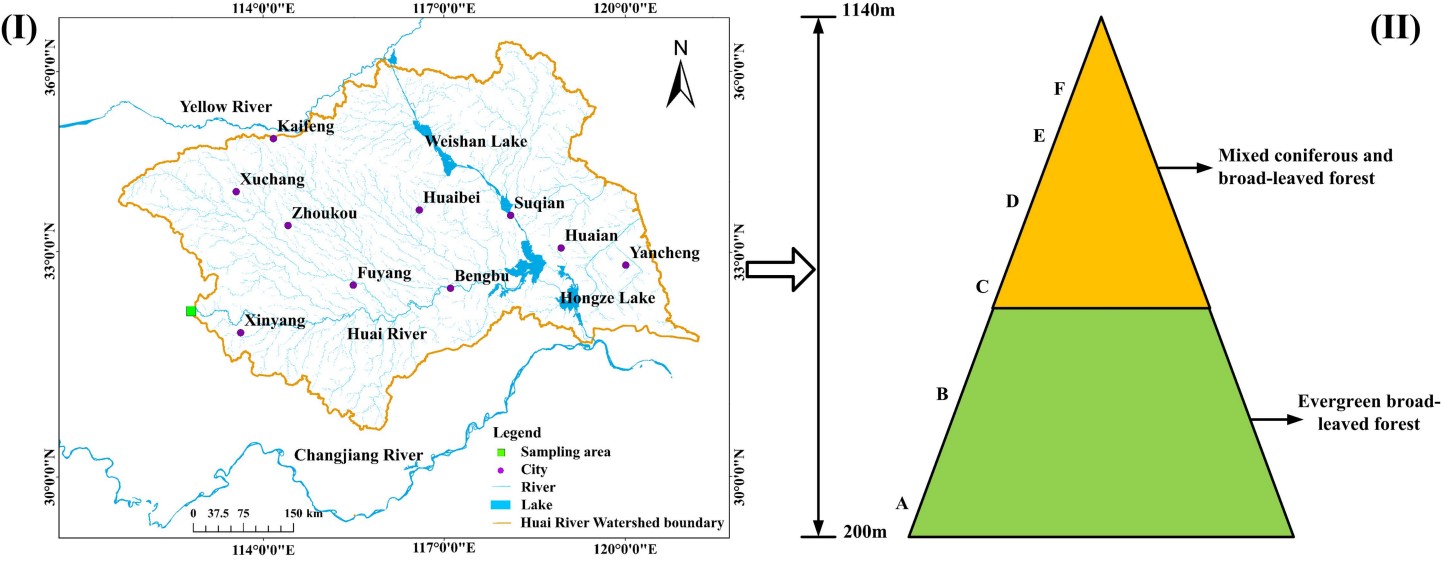

Fig 1. (I) Geographical map displaying the location of the study area, and (II) Schematic diagram of vertical vegetation zonation along an altitudinal gradient in Tongbai mountain (A-F).

Table 1. Sampling locations with latitude, longitude, and elevation.

| Sampling locations | Latitude (N) | Longitude (E) | Elevation (m) |
|---|---|---|---|
| A | 32°24'5.73" | 113°17'6.56" | 288 |
| B | 32°23'49.40" | 113°17'18.08" | 485 |
| C | 32°23'35.02" | 113°17'4.24" | 641 |
| D | 32°23'13.50" | 113°16'52.62" | 823 |
| E | 32°22'59.56" | 113°16'43.02" | 927 |
| F | 32°22'54.91" | 113°16'43.54" | 991 |

treated samples were used for comprehensive characterisation using various analytical techniques such as SEM, FTIR, XRD, and XRF to reveal the mineral composition, SOC characteristics, and elemental distribution of the soil.

## 2.2. XRD analysis of soil minerals

XRD technology is used to analyse soil minerals. XRD technology has a wide range of advantages in terms of its efficient, non-destructive, and quantitative analysis of the type, content, and crystal structure of soil minerals. It is particularly suitable for the study of complex mineral systems [26]. The treated sample was made into a powder wafer and analysed using a Rigaku Smartlab9 fully automatic X-ray diffractometer. Determination conditions included: Cu-Kα target; graphite monochromator filter; tube pressure: 40kV; current: 150 mA; scanning range: 5°∼80° (2θ); scanning speed: 6°min$^{-1}$ (2θ). The diffraction peaks obtained by XRD analysis were processed using the Jade software, and the diffraction peak results were compared with the standard PDF card to obtain phase analysis results. According to the adiabatic method, the integral intensities of selected diffraction peaks of clay minerals and various non-clay minerals in the diffraction pattern were measured, and the total amount of clay minerals and the content of non-clay minerals were directly calculated using the following formula:

$$X_i = \left[ \frac{\frac{I_i}{K_i}}{\left( \sum \frac{I_i}{K_i} \right)} \right] \times 100\%$$

where $X_i$ is the percentage content of mineral $i$ in the sample, expressed as a percentage; $K_i$ is the reference intensity of mineral $i$; and $I_i$ is the intensity of a certain diffraction peak of mineral $i$.

## 2.3. SEM analysis of aggregates

The structure of soil microaggregates was then studied using a scanning electron microscope (Zeiss GeminiSEM 360, ZEISS, Germany). Soil powder was adhered to a conductive adhesive during the experiment and tested after gold sputtering using an ion sputter [27].

## 2.4. FTIR analysis of soil organic molecules

In this study, FTIR was used to analyse the composition and distribution of SOC and its functional groups [28,29]. The samples used were naturally air-dried soil samples, which were processed through a 2 mm sieve, mixed with spectrally pure potassium bromide at a mass ratio of 1:90, ground manually in an agate mortar, and pressed into tablets. FTIR analysis was performed using a Thermo Nicolet iS5 Fourier transform infrared spectrometer (ThermoFisher Scientific, USA).

## 2.5. XRF analysis of soil elemental compounds

XRF was selected as the analytical technique in this study because it can retain the original structure of the soil and quickly achieve high-precision quantitative analysis of multiple elements. It is especially suitable for the comprehensive characterisation of heavy metals and conventional elements in complex soil matrices [30]. The dried soil sample was ground to 200 mesh, and the powder sample was weighed and pressed into a thin sheet, which was then placed in an X-ray fluorescence spectrometer for elemental analysis. The dried soil sample was ground to 200 mesh, and the powder sample was weighed and pressed into a thin sheet, which was then placed in an X-ray fluorescence spectrometer for elemental analysis. The XRF analysis was performed using an AXIOS (PW4400) wavelength dispersive X-ray fluorescence spectrometer (PANalytical B.V., Almelo, the Netherlands), with a maximum power of 4.0 kW, a maximum excitation voltage of 60kV, and a maximum current of 125mA. Equipped with an SST ultra-sharp ceramic end-window (75μm), a Rh target X-ray tube, and a 68-position (32 mm diameter) sample changer, the system utilizes SuperQ 4.0 quantitative analysis software. The software

automatically performs peak intensity calculations, Compton scatter internal standard correction, spectral interference correction, and matrix correction, and then displays the analysis results. At the same time, XRF quantitative analysis provided information on the content of MgO, $P_2O_5$, $K_2O$, and CaO. Calibration was carried out using certified reference materials (GBW07401 and GBW07404 from the Chinese National Institute of Metrology), which cover the major and trace element concentration ranges relevant to soil. The analytical accuracy was validated through repeated testing of these reference materials, with the results compared to their certified values, yielding recovery rates typically ranging from 90% to 110%.

The limits of detection (<LOD) for each element were determined using the $3\sigma$ criterion (three times the standard deviation of blank measurements) and were incorporated into subsequent quantitative analyses. The analysis provided quantitative data for major oxides (MgO, $P_2O_5$, $K_2O$, CaO) as well as additional elements such as Fe, Na, and Ba. For instance, a reported value of "<LOD" indicates that the element's concentration is below the quantification limit, with the uncertainty representing the standard deviation from replicate analyses. Although iron (Fe) was determined with high accuracy, its data were treated separately from the major oxides due to its distinct concentration range and specific relevance to the study objectives. Similarly, high relative standard deviations observed for elements like barium (Ba) (e.g., 435 ± 399) likely reflect low concentration levels and inherent challenges associated with spectral interferences in complex soil matrices. Future investigations will focus on further refining these measurements to minimize uncertainty.

## 2.6. Data statistics, modelling, and mapping

This study used a multi-faceted approach to analyse correlations between altitude and the elemental, oxide, mineral, and FTIR spectral data. A comprehensive feature scoring method was constructed, combining Pearson correlation coefficients between the feature and altitude, random forest feature importance, and principal component analysis (PCA) loadings to form a comprehensive scoring formula.

The comprehensive scoring formula was.

$$Comprehensive\ score\ =\ |\ correlation\ coefficient\ |\ +\ |\ random\ forest\ importance\ |\ +\ |\ PCA\ loadings\ |$$

This method takes into account linear correlations, non-linear predictive power, and the contribution of features to overall data variability, overcoming the limitations of single-indicator assessments. By combining multiple analytical methods, the importance of features was comprehensively assessed, the accuracy and reliability of sensitive factor screening were improved, the impact mechanism of altitude gradient on soil biogeochemical characteristics was deeply analysed. As a result, a new scientific method was developed to understand the ecological processes in the vertical zonation of Mount Tongbai. Statistical analyses were performed using SPSS 15.0 for Windows (SPSS Inc, Chicago, IL). The map was generated using ArcGIS 10.0 (ESRI, Redlands, CA, USA: http://www.esri.com/software/arcgis). All connecting lines were generated using the Origin 8.1 Software. (Origin Lab Corp. v 8.1).

## 3. Result and analysis

### 3.1. Gradient changes in mineral composition and physicochemical properties with altitude

The XRD analysis results showed that the mineral composition of the Tongbai Mountain soil exhibited a clear gradient change with altitude (Fig 2,3; Table 2). At the lowest altitude point A and the highest altitude point F, the Quartz (low) content was high, at 55.8% and 56.3% respectively, showing a U-shaped distribution. This fphenomenon may be related to the degree of weathering and the material source [16]. The content of microcline reached a peak at the mid-altitude point C, at 29.7%, and then decreased significantly to 5.5% at the high-altitude point F. The content of albite was highest at mid-altitude site B, at 34.0%, and then gradually decreased with increasing altitude. This may reflect the fact that feldspar minerals are more stable at mid-altitude. Illite increased with increasing altitude (Fig 2,3; Table 2). Kaolinite was more abundant at mid-to-low altitudes. The content of clay minerals such as chlorite did not change much at different altitudes.

The results of the XRF analysis showed that the content of the main chemical components of the soil varied significantly with altitude (Fig 4; Table 3). The $Al_2O_3$ content was higher at mid-altitude points B, C, and D, ranging from 18.42% to 20.04%, which may be related to weathering of feldspar minerals and aluminium enrichment. The MgO content at the low-altitude sites A, B, and C was < LOD, indicating values below the limit of detection; while at the high-altitude sites D, E, and F, it increased significantly up to 2.47%. This may be related to the contribution of magnesium-containing minerals in the parent rock and the vegetation type at high altitudes. Magnesium is an essential nutrient for plants, participating in various physiological and biochemical processes, and also plays an important role in enzyme activation, protein synthesis, and carbohydrate metabolism [31]. The $K_2O$ content gradually increased from point A to point C, and then gradually decreased at higher altitudes, which was consistent with the trend of the content of plagioclase feldspar. Plagioclase feldspar is rich in K and may be an important source of $K_2O$. This indicates that the content of plagioclase directly affects the potassium content in the soil.

SEM analysis revealed the microscopic characteristics of soil particles (Fig 5,6). Soil microscopic structure analysis showed that as the altitude increased, the morphology and aggregation state of soil particles changed significantly. In the low altitude samples A and B, the soil microaggregates were arranged, the particle size distribution was relatively uniform, and the porosity was high. In the middle-altitude samples C and D, obvious agglomerate structures began to appear, and the bonding between particles was enhanced. In the high-altitude samples E and F, a more complex microstructure could be observed, including laminar arrangements and a well-developed pore network. These changes may be related to the higher SOC and clay mineral content at high altitudes, which promote the formation and stability of soil aggregates [32].

## 3.2. Altitudinal gradient characteristics of SOC and changes in functional groups

SOC is a key component of soil ecosystems that affects soil fertility and the carbon cycle [19]. Changes in soil carbon storage are obvious along vertical gradients [33]. In this study, FTIR was used to semi-quantitatively analyse the organic components in soil samples from different altitude zones and to explore the distribution patterns of different organic groups in the environment

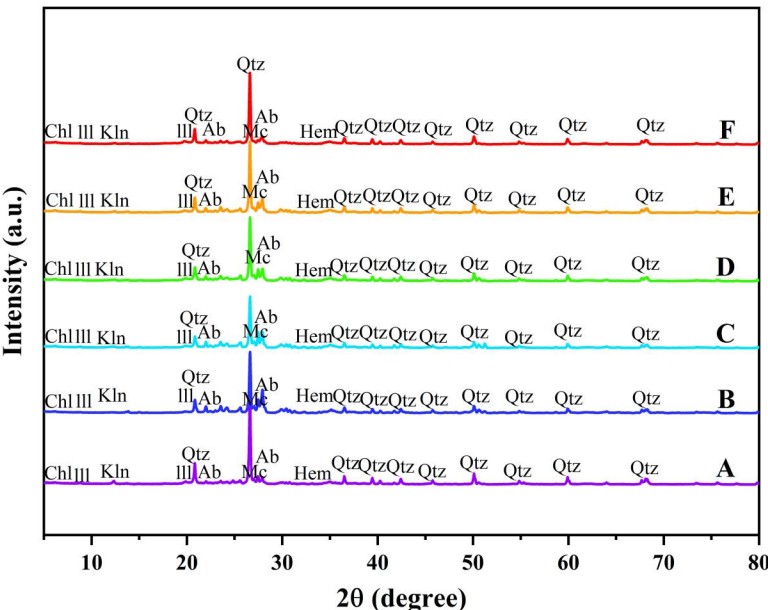

**Fig 2. X.ray diffractogram patterns (CuKα radiation) and mineral change of soil aggregates across sampling sites (Mc = Microcline; Kln = Kaolinite; Ill = Illite; Qtz = Quartz (low); Chl = Chlorite; Ab = Albite; Hem = Hematite.).**

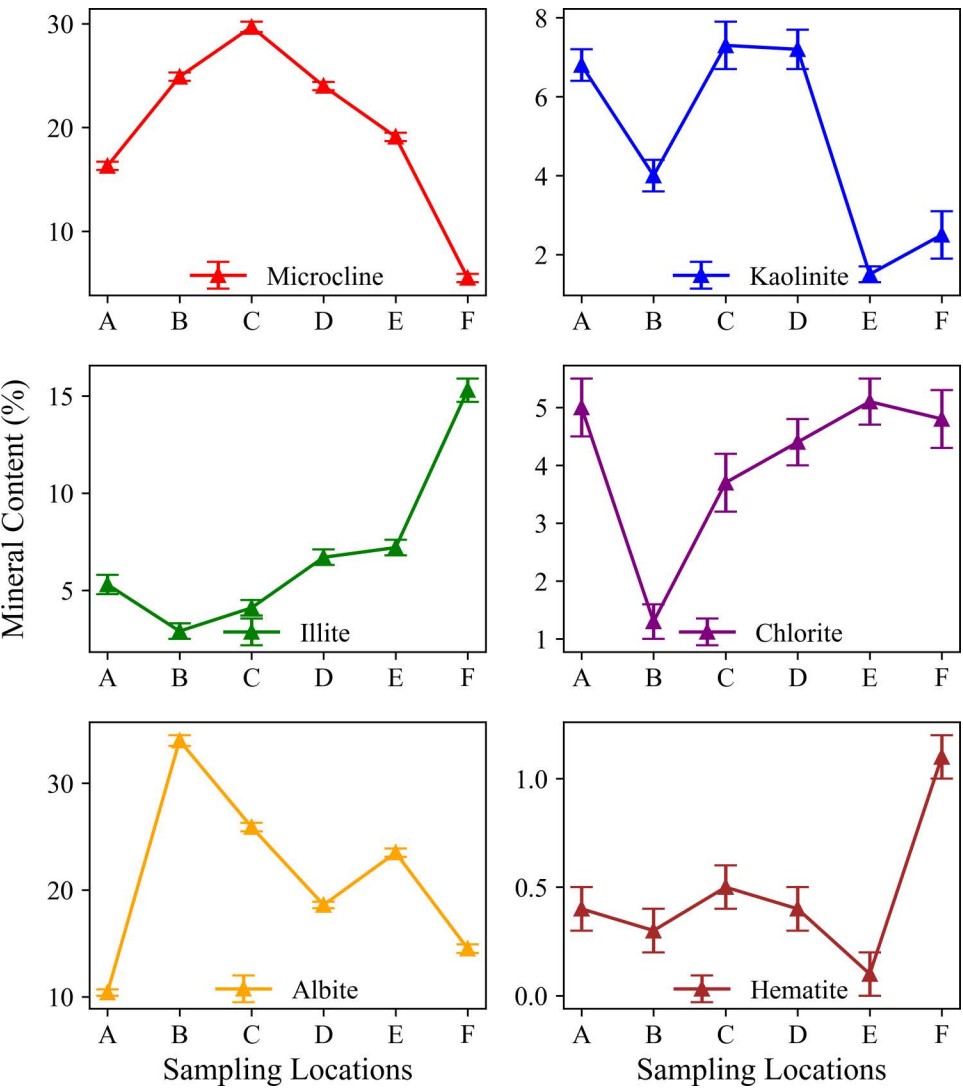

**Fig 3. Mineral content distribution across sampling locations (A-F).**

[29]. The FTIR analysis results showed that the SOC content increased with increasing altitude, reaching a maximum at the highest altitude point F (Fig 7; Table 4) [5]. This trend may be related to the lower temperature, higher precipitation, and special vegetation types observed at high altitudes, which collectively promote the accumulation and preservation of SOC [34]. However, other studies have found different patterns. Yang's [35] study in the Cangshan National Nature Reserve in the Yunnan Province showed that mid-altitude areas were conducive to the accumulation of SOC. A clear absorption peak was also observed at 3620 cm⁻¹, which indicated the presence of mineral hydroxyl groups (Fig 7; Table 4) [36]. This peak was enhanced at high altitudes, which may be related to the high content of mineralised organic matter. In this study, the functional group type of SOC also showed a clear altitude gradient. The peak at 3423 cm⁻¹ corresponds to -OH or -NH groups. The stronger and broader peak at 3423 cm⁻¹ indicates the presence of more hydroxyl compounds in the sample, such as humus or organic acids [12,37]. The absorption peak at 2929 cm⁻¹ represents aliphatic C-H stretching vibrations and generally indicates the presence of long-chain fatty acids or hydrocarbons in SOC [29,38]. The relative intensity of these absorption peaks was significantly

**Table 2. Mass fraction of soil mineral using XRD with standard deviation (%).**

| Mineral (%) | A | B | C | D | E | F |
|---|---|---|---|---|---|---|
| Microcline | 16.3±0.4 | 24.9±0.4 | 29.7±0.5 | 24.0±0.4 | 19.1±0.4 | 5.5±0.4 |
| Kaolinite | 6.8±0.4 | 4.0±0.4 | 7.3±0.6 | 7.2±0.5 | 1.5±0.2 | 2.5±0.6 |
| Illite | 5.3±0.5 | 2.9±0.4 | 4.1±0.4 | 6.7±0.4 | 7.2±0.4 | 15.3±0.6 |
| Quartz (low) | 55.8±0.7 | 32.6±0.4 | 28.8±0.4 | 38.8±0.4 | 43.5±0.4 | 56.3±0.8 |
| Chlorite | 5.0±0.5 | 1.3±0.3 | 3.7±0.5 | 4.4±0.4 | 5.1±0.4 | 4.8±0.5 |
| Albite | 10.4±0.3 | 34.0±0.5 | 25.9±0.4 | 18.6±0.3 | 23.5±0.4 | 14.5±0.4 |
| Hematite | 0.4±0.1 | 0.3±0.1 | 0.5±0.1 | 0.4±0.1 | 0.1±0.1 | 1.1±0.1 |

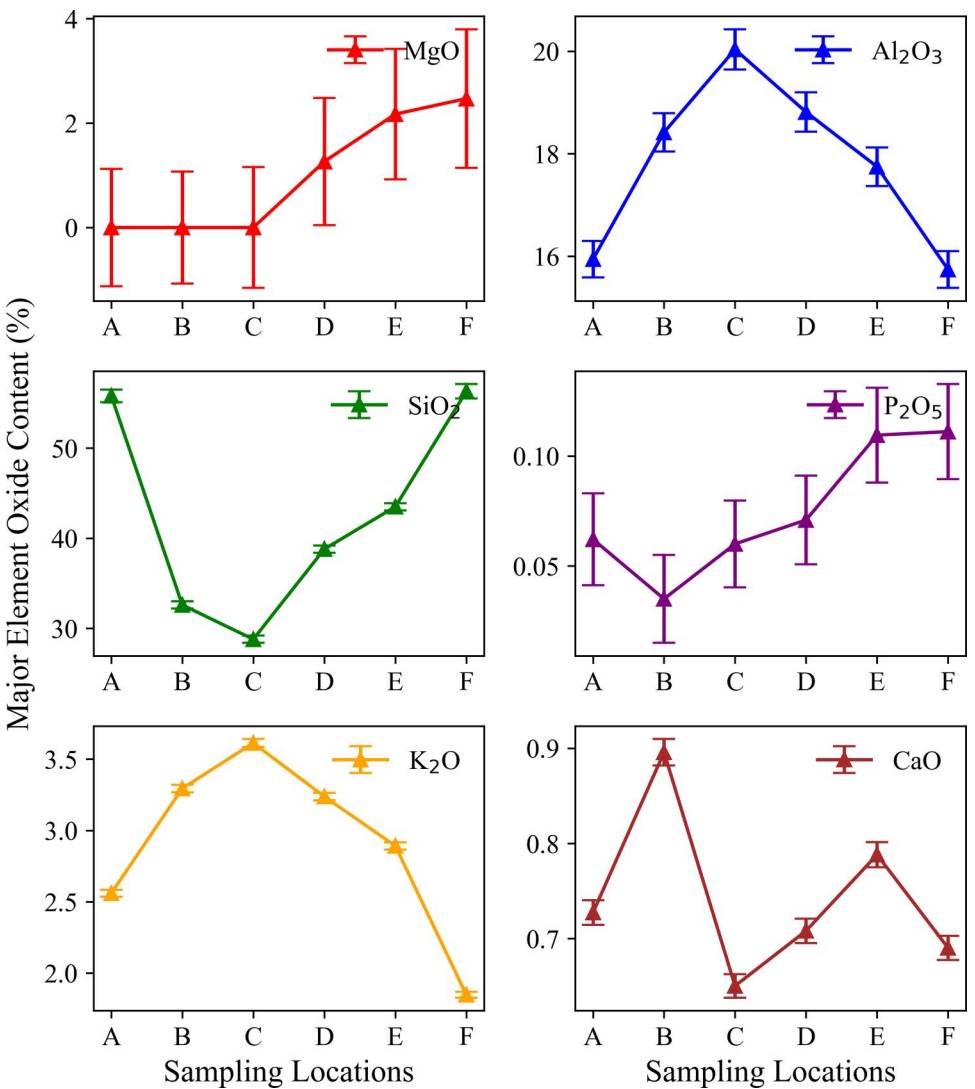

**Fig 4. Major element oxide content variation across different sampling locations.**

**Table 3. Major element oxide concentrations at different sampling locations determined by XRF and <LOD indicating below the limit of detection.**

| Compounds (%) | A | B | C | D | E | F |
|---|---|---|---|---|---|---|
| MgO | <LOD | <LOD | <LOD | 1.26±1.22 | 2.17±1.25 | 2.47±1.33 |
| $Al_2O_3$ | 15.94±0.36 | 18.42±0.37 | 20.04±0.39 | 18.81±0.38 | 17.75±0.38 | 15.74±0.36 |
| $SiO_2$ | 55.80±0.70 | 32.60±0.40 | 28.80±0.40 | 38.80±0.40 | 43.50±0.40 | 56.30±0.80 |
| $P_2O_5$ | 0.06±0.02 | 0.04±0.02 | 0.06±0.02 | 0.07±0.02 | 0.11±0.02 | 0.11±0.02 |
| $K_2O$ | 2.56±0.02 | 3.29±0.03 | 3.61±0.03 | 3.24±0.03 | 2.89±0.03 | 1.85±0.02 |
| CaO | 0.73±0.01 | 0.90±0.01 | 0.65±0.01 | 0.71±0.01 | 0.79±0.01 | 0.69±0.01 |

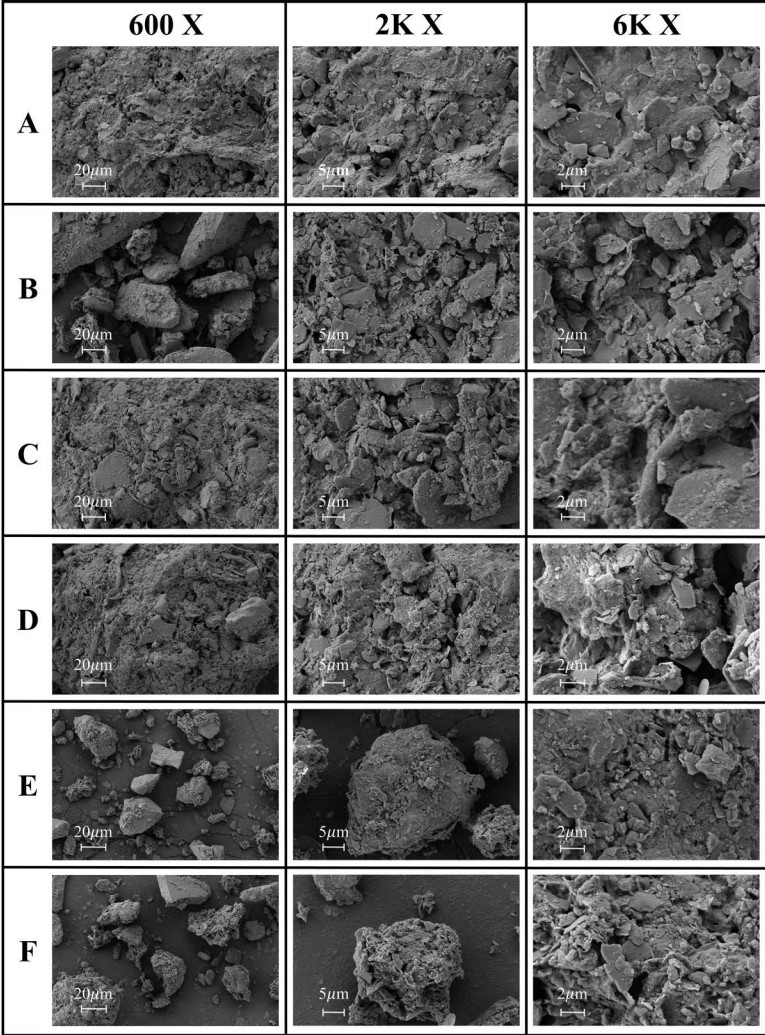

**Fig 5. SEM photographs of soil particles at 600×, 2K × and 6K×magnification across different sampling locations.**

enhanced at high altitude points E and F, reaching 49.97 and 5.88 at 3423 cm⁻¹ and 2929 cm⁻¹, respectively (Fig 8; Table 4), indicating that high-altitude soils contained more hydroxyl, amino, and aliphatic compounds. The absorption peak at 1631 cm⁻¹ usually reflects the presence of C=O or C=C and may be related to the accumulation of humic substances, fulvic acids, or

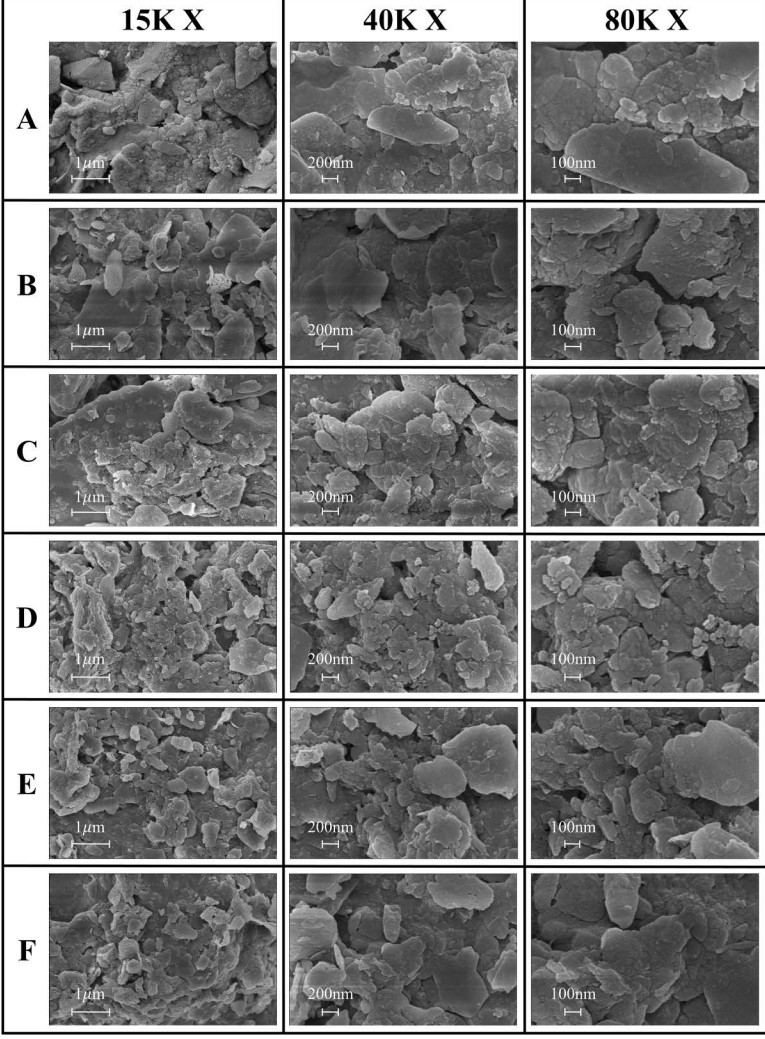

**Fig 6. SEM photographs of soil particles at 15K×, 40K × and 80K×magnification across different sampling locations.**

aromatic compounds [29]. The 1631 cm⁻¹ absorption peak reached a maximum value of 15.23% at the mid-elevation D site, reflecting the enrichment of aromatic SOC at this elevation [39]. This may be related to the specific vegetation type and microbial activity in the middle-elevation area, which promotes the accumulation of aromatic compounds. Functional groups such as hydroxyl groups (3423 cm⁻¹) and aliphatic hydrocarbons (2929 cm⁻¹) in SOC are important during the decomposition and transformation of SOC and can reflect components in the soil that are more easily decomposed [40].

In order to more accurately assess the amount of SOC, research methods from Gerzabek, Antil (41) were used. They found that absorption peaks at 2920 cm⁻¹, 1630 cm⁻¹, and 1450 cm⁻¹ were closely related to the SOC content [41]. Although the 1450 cm⁻¹ peak was not detected in this study, a similar absorption peak was observed at a similar wave number (e.g., 1631 cm⁻¹). Based on these observations, a SOC-FTIR index was constructed to comprehensively measure the SOC content. The index was obtained by summing the normalized intensities of absorption peaks at 3620 cm⁻¹, 3423 cm⁻¹, 2929 cm⁻¹, and 1631 cm⁻¹. The value of the F point was up to 72.74, and the SOC content in the high-altitude area on the surface was relatively high (Table 4). This method took into account differences in absorbance of different functional groups and aimed to provide a more comprehensive indicator to evaluate SOC. Altitude change is an important factor affecting SOC and its stabilization process. The low temperatures at higher altitudes slow down the decomposition rate of SOC and promote its accumulation [17]. In addition, the special vegetation

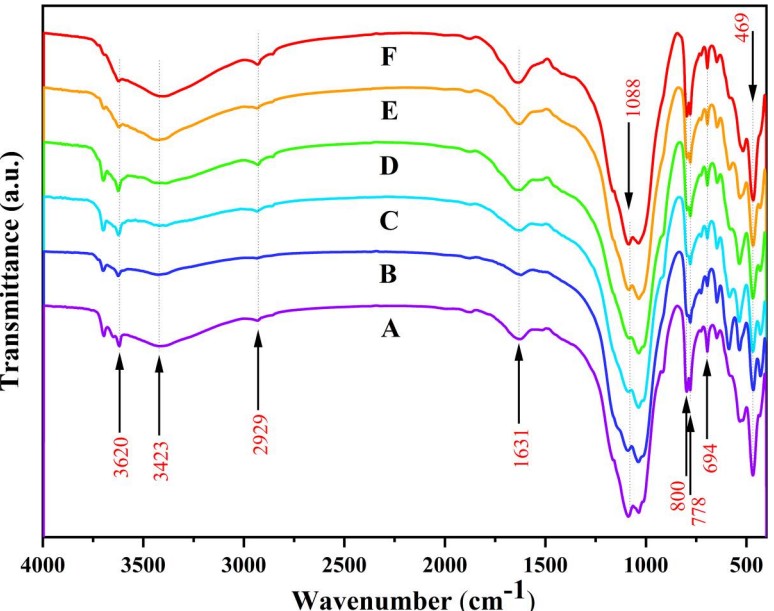

**Fig 7. Infrared spectrum characteristics of soil of different sampling locations.**

**Table 4. Integral of relative peak area of main peaks of FTIR spectrum.**

| λ/cm⁻¹ | A | B | C | D | E | F |
|---|---|---|---|---|---|---|
| SOC-FTIR index SUM | 42.28±0.04 | 23.39±0.19 | 35.91±0.12 | 52.98±0.18 | 53.08±0.01 | 72.74±0.05 |
| 3620 | 1.93±0.01 | 1.13±0.03 | 1.85±0.00 | 2.42±0.00 | 2.06±0.00 | 2.60±0.01 |
| 3423 | 28.49±0.03 | 14.46±0.09 | 21.02±0.05 | 31.38±0.10 | 37.49±0.01 | 49.97±0.02 |
| 2929 | 2.66±0.01 | 1.15±0.03 | 2.56±0.02 | 3.95±0.04 | 3.96±0.00 | 5.88±0.01 |
| 1631 | 9.20±0.03 | 6.65±0.16 | 10.48±0.11 | 15.23±0.15 | 9.57±0.01 | 14.29±0.04 |
| 1088 | 46.12±0.32 | 44.19±0.25 | 41.63±0.34 | 41.30±0.33 | 42.88±0.32 | 46.31±0.35 |
| 800 | 3.59±0.06 | 2.48±0.09 | 2.23±0.09 | 2.64±0.04 | 2.79±0.06 | 3.45±0.07 |
| 778 | 4.55±0.01 | 4.78±0.03 | 4.54±0.07 | 4.73±0.07 | 4.61±0.07 | 4.18±0.08 |
| 694 | 1.95±0.01 | 1.55±0.00 | 2.07±0.02 | 1.93±0.05 | 1.61±0.02 | 1.31±0.02 |
| 469 | 17.19±0.02 | 12.33±0.10 | 14.86±0.11 | 15.23±0.12 | 15.75±0.01 | 17.30±0.11 |

types and microbial community structures at high altitudes also have an impact on the accumulation and transformation of SOC. SOC derived from microbial residues, such as enzymes, exopolysaccharides, and lipids, is more stable in soil. Particularly at high altitudes and low temperatures, the metabolic rate of microorganisms is reduced, and SOC is more preserved [12,17]. The soil at high altitudes had a higher content of hydroxyl, amino, and aliphatic compounds, while the middle-altitude area had an aromatic SOC enrichment (Table 4). These results indicate that factors such as climatic conditions and microbial activity jointly affect the composition and distribution of SOC.

### 3.3. Spatial distribution of heavy metal elements and soil microstructure characteristics

This study found that the distribution of heavy metal elements in mountain soil showed a significant altitude gradient effect, reflecting the complexity of element migration and accumulation in mountain ecosystems [2]. Analysis of the content of major elements showed that different elements exhibited unique distribution patterns with altitude. The content of Fe and Ti elements increased with increasing altitude (Fig 9; Table 5). At the low altitude point A, the Fe content was 24280 mg·kg⁻¹, and the Ti content was

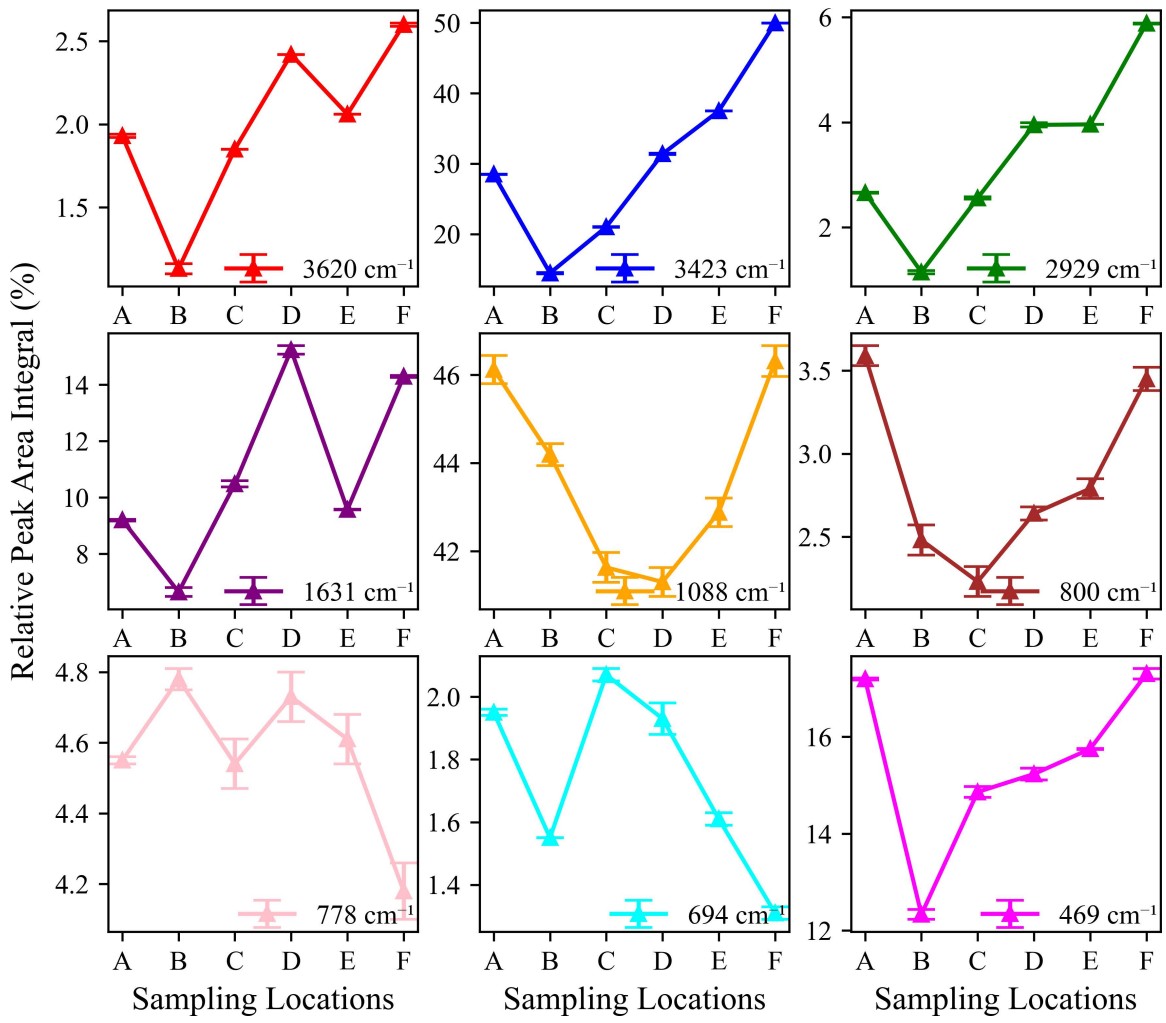

**Fig 8. Relative FTIR peak area integral across sampling locations.**

3462 mg·kg⁻¹. At the high-altitude site F, the Fe content increased to 32943 mg·kg⁻¹, and the Ti content increased to 4063 mg·kg⁻¹. Cu and Zn elements also showed a slight increasing trend with increasing altitude. The Cu content increased from 34 mg·kg⁻¹ at the low-altitude site A to 42 mg·kg⁻¹ at the high-altitude site F, and the Zn content increased from 119 mg·kg⁻¹ to 172 mg ·kg⁻¹. The high Fe, Ti, Cu, and Zn concentrations seen at high altitudes may be related to the higher SOC content, which enhances the adsorption capacity of the soil for these trace elements [2,33,42]. Sr was enriched at medium altitude (Fig 9; Table 5). At the middle altitude point B, the Sr content reached a maximum value of 668 mg·kg⁻¹, while at other altitude points, its content was below 500 mg·kg⁻¹. This may be related to the specific geological background of the area, where parent rocks are rich in Sr-bearing minerals, which are weathered and release Sr into the soil [43]. The distribution of V element is also of interest. In this study, the correlation coefficient between V, Fe, and Cu was 0.9, indicating a strong positive correlation (Fig 10). The correlation coefficient between V and Mn was 0.7, indicating a moderate positive correlation. This indicates that the migration and enrichment of V are controlled by Fe, Cu, and Mn oxides, which may have similar behaviour during redox reactions and element migration. Related studies have shown that Fe and Mn may promote the adsorption and fixation of V [44]. Element correlation analysis showed that Fe was significantly positively correlated with V, Cu, and Zn (r>0.7), indicating that these elements may have similar geochemical

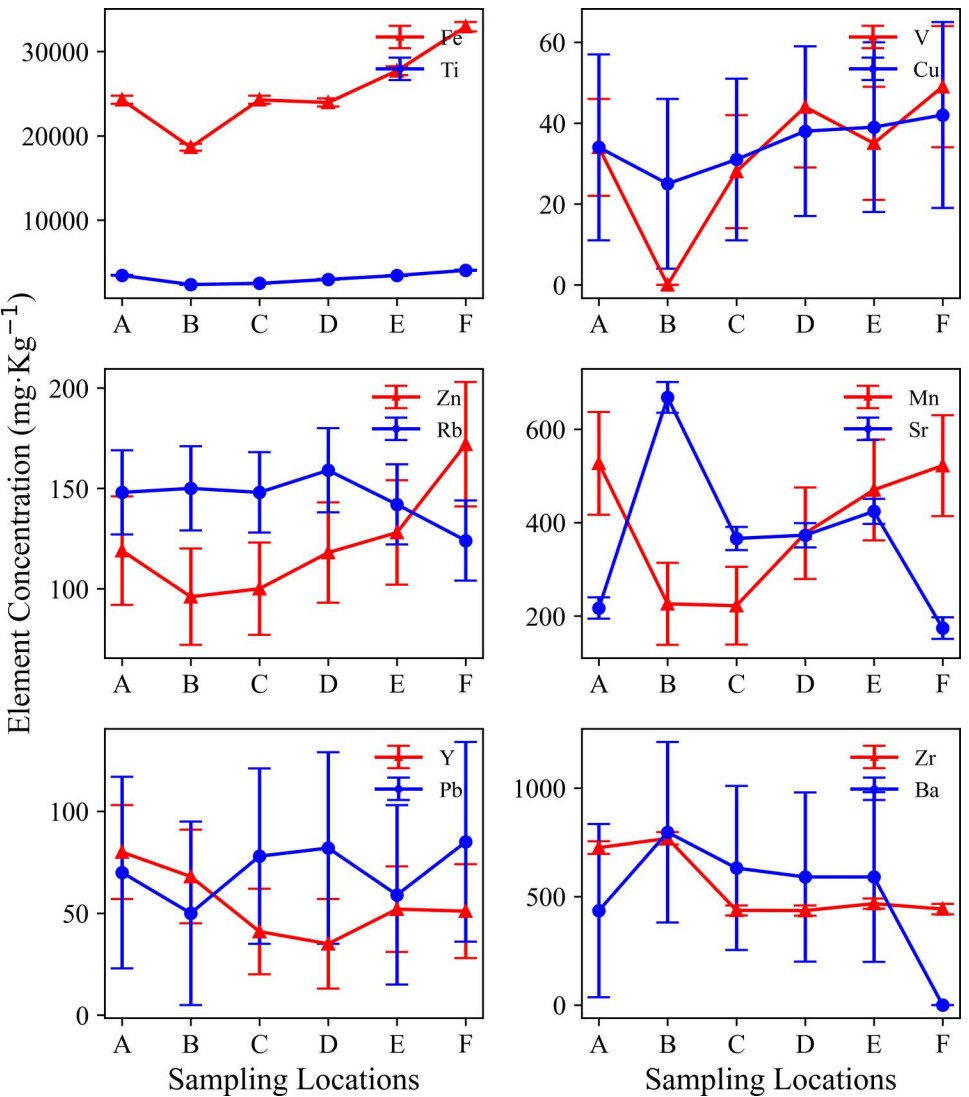

**Fig 9. Elemental concentration variations across sampling locations.**

behaviours and a common source [45], mainly from parent rock weathering and soil formation processes (Fig 10 and 11). Sr was strongly positively correlated with Ba, but negatively correlated with Fe and Ti [46]. In high-altitude samples, more iron oxides and clay minerals are attached to the surface of soil particles, and these minerals have a strong ability to adsorb heavy metals [14,32]. The correlation analysis used the Pearson correlation coefficient, and there was a significant correlation between multiple elements, supporting the hypothesis that the elements migrated and were enriched together (Fig 9).

## 4. Discussion

### 4.1. Distribution patterns of minerals, SOC, and heavy metal elements and their altitude gradient effects

This study revealed the collaborative evolution process of soil mineral composition, SOC characteristics, and heavy metal element distribution in the vertical belt spectrum of Mount Tongbai, which was significantly driven by the altitude gradient. The Quartz content was higher in the low and high-altitude areas, and Quartz was preserved as a resistant mineral (Fig

**Table 5. Mass fraction of soil heavy elements determined by XRF with standard deviation (mg·Kg⁻) and <LOD indicating below the limit of detection.**

| Element (mg·Kg-1) | A | B | C | D | E | F |
|---|---|---|---|---|---|---|
| Fe | 24280±480 | 18649±423 | 24267±481 | 23948±478 | 27718±514 | 32943±557 |
| Ti | 3462±51 | 2375±42 | 2515±44 | 2989±48 | 3446±51 | 4063±57 |
| V | 34±12 | <LOD | 28±14 | 44±15 | 35±14 | 49±15 |
| Cu | 34±23 | 25±21 | 31±20 | 38±21 | 39±21 | 42±23 |
| Zn | 119±27 | 96±24 | 100±23 | 118±25 | 128±26 | 172±31 |
| Rb | 148±21 | 150±21 | 148±20 | 159±21 | 142±20 | 124±20 |
| Mn | 527±110 | 226±88 | 222±83 | 377±98 | 470±108 | 522±108 |
| Sr | 217±23 | 668±33 | 366±25 | 373±26 | 424±27 | 174±23 |
| Y | 80±23 | 68±23 | 41±21 | 35±22 | 52±21 | 51±23 |
| Pb | 70±47 | 50±45 | 78±43 | 82±47 | 59±44 | 85±49 |
| Zr | 725±29 | 768±29 | 436±23 | 435±24 | 467±24 | 442±24 |
| Ba | 435±399 | 796±416 | 631±378 | 590±390 | 590±391 | <LOD |

2,3; Table 2). However, the Quartz content decreased in the middle altitude area. Feldspar-like minerals (e.g., microcline and albite) had the highest content in the middle-elevation area, which may be related to moderate climatic conditions [16]. The higher Al2O3 content seen in the middle-elevation area suggests an enrichment of clay minerals (Fig 4; Table 3), which may function as a cementing agent to promote particle aggregation (Fig 4; Table 3) [47]. Chlorite and illite were found in higher concentrations at point F (Fig 3; Table 2). A study on highland soils in Hunan, China, demonstrated that in subtropical mountainous regions, the proportion of kaolinite decreases while the contents of illite and vermiculite increase with rising elevation, reflecting a reduction in weathering under high-altitude, low-temperature conditions [48]. Our study also revealed that with increasing elevation, the kaolinite content in soils decreased and the illite content increased, which is consistent with these findings and provides empirical support for the mechanisms of soil evolution in subtropical mountainous areas. As the altitude increased, the weathering of feldspar minerals released more clay minerals to form a complex structure of microaggregates. There was a significant positive correlation between chlorite, illite, and SOC (Fig 10) [49]. Chlorite may adsorb more low molecular weight organic matter through its surface sites at high altitudes, while illite may better protect SOC through a more stable complex structure (Fig 5,6) [5,50–52]. The variation of SOC charac- teristics with altitude showed a clear vertical zonation (Fig 7; Table 4). The temperature at point F, calculated from the difference in elevation, was likely 4.57°C lower than that at point A. The higher SOC content at point F in high-altitude areas was related to lower temperatures and higher precipitation (Table 4). These conditions inhibit the decomposition of SOC and promote its accumulation. Due to global warming, a temperature rise is likely to exacerbate the decomposition of SOC. Similar patterns have also been observed in other mountainous regions. For instance, in the subtropical Lushan Mountains of China, studies have indicated that SOC stocks in high-altitude areas (1268 m) are significantly higher than those in low-altitude areas, a trend closely linked to increased annual precipitation and decreased annual temperature [53]. In central China's Dabie Mountains—a transitional zone between northern and southern climates—a 2024 study on subtropical coniferous forests found that soils at higher elevations exhibited both higher SOC contents and more stable aggregate structures. These results suggest that cooler and more humid conditions retard organic matter decomposition, thereby promoting SOC accumulation and enhancing aggregate stability, which in turn results in soils with richer and more robust organic matter at higher altitudes [54]. Moreover, in the Bamtou Mountains of Cameroon, Central Africa, SOC reserves were found to increase with altitude, reaching a peak at 2740 m [5], while research in the Karakorum Mountains also demonstrated a significant positive correlation between SOC and altitude [55]. In addition, the increase in -OH and -NH groups reflected special moisture conditions and nitrogen cycle processes at high altitudes (Fig 7; Table 4) [56]. The

**Fig 10.** Correlation heatmap of elemental concentrations, mineral contents, and FTIR spectra for soil samples, the different colors in the heatmap represent the correlation strength between variables, with red indicating a strong positive correlation, blue indicating a strong negative correlation, and lighter colors representing weaker correlations.

wavenumbers $3620\,cm^{-1}$, $3423\,cm^{-1}$, $2929\,cm^{-1}$, and $1631\,cm^{-1}$ were highly correlated with V, Cu, Zn, and Fe, respectively, with correlation coefficients greater than 0.6. A high positive correlation with Pb and Mn was also seen. These elements were also highly correlated with illite and chlorite. The content of heavy metal elements such as Fe, Ti, Cu, and Zn increased with altitude, which may be due to the higher content of secondary clay minerals such as illite and chlorite, and

the high SOC content at high altitudes, which have a strong adsorption and fixation effect on heavy metal elements (Fig 9; Table 5) [2,32]. Research on the southern slope of Mount Everest has revealed that the concentrations of Co, Zn, Cr, and Ni increase with elevation, reaching a peak between 4900 and 5000 meters [57]. Similarly, studies in the Alps have shown that Pb, Cd, and Zn concentrations rise with altitude, primarily due to long-range atmospheric transport [58]. Additionally, a study conducted on Luoji Mountain in Southwest China (2200–3850 m) demonstrated that Cd levels in organic-rich soils at high altitudes are significantly elevated, and lead isotope analysis indicated that Pb in the high-altitude organic horizon mainly originates from anthropogenic emissions, accounting for over 65% of the total [59]. These findings suggest that the "cold trapping" effect in high-altitude, cold climates may facilitate the accumulation of certain heavy metals with increasing elevation. However, although our study observed higher concentrations of elements such as Fe, Ti, Cu, Zn, and Pb in high-altitude soils, it remains to be determined whether these increases are directly attributable to the cold trapping effect. Furthermore, Sun et al. [2] reported that in the high-mountain soils of the Hengduan Mountains in Southwest China, climatic factors—such as mean annual temperature and precipitation—in conjunction with soil organic matter and other properties collectively regulate the spatial distribution of elements such as Fe, Mn, Zn, and Cu [2]. This evidence confirms that vertical altitude gradients profoundly shape the geochemical characteristics of mountainous soils, resulting in significant differences in elemental distribution across various elevations. The findings of our study further corroborate the regulatory effect of altitude gradients on the distribution of soil trace elements. The low temperature and high humidity conditions at high altitudes, as well as the high SOC and clay mineral content, increase the SOC content and further promote the adsorption and fixation capacity of the soil for some heavy metal elements [32]. Zr is more abundant at lower altitudes and is used as a stable marker to track soil erosion and weathering processes due to its ability to resist environmental changes [60].

MgO was only detected at higher-altitude sites (Fig 4; Table 3). In steep mountain terrain, MgO is prone to migration with water flow and infiltration to depth, leading to the vertical loss of magnesium ions. Mg has a significant effect on soil microbial biomass, activity, and bacterial community composition [61]. The regulation of heavy metal behaviour by SOC can also indirectly affect ecosystem function by influencing the structure and activity of microbial communities. Organo-metallic complexes may provide a living environment for specific microorganisms such as heavy metal tolerant bacteria, which are involved in the transformation and detoxification of heavy metals, thereby affecting nutrient cycling and ecosystem productivity [62].The complex microstructure, higher porosity, and abundant SOC content in high-altitude areas collectively enhance the soil's water retention capacity, thereby promoting vegetation growth and ecosystem stability (Fig 5–7; Table 4) [63]. In addition, a good aggregate structure also improves the soil's erosion resistance, which is essential to maintain the stability of mountain ecosystems [64]. Microaggregate structures promote nutrient mineralisation and cycling by providing microhabitats for microorganisms, and the organic matter secreted by microorganisms forms organic-mineral complexes on the surfaces of soil particles, which further enhances the formation of microaggregates [65,66]. The complex evolution of the structural functions of microaggregates may form a positive feedback relationship with microorganisms [67,68]. A study conducted in the semi-arid Helan Mountains of northwest China (elevations 1400–2400 m) demonstrated that the proportion of SOC within aggregates increases significantly with elevation. Moreover, in high-altitude areas, SOC is primarily stabilized through physical protection provided by clay minerals within these aggregates [69]. The reproduction of microorganisms promotes the increase of SOC, which further promotes the formation of clay minerals and the stability of soil aggregates [7,9,70]. The binding of minerals to SOC not only strengthens the structure of microaggregates, but also enhances their persistence by providing reactive surfaces [11]. From an ecosystem service perspective, the interaction of soil structure, minerals, and SOC along an elevation gradient affects water conservation, carbon sequestration, and biodiversity maintenance [71]. Better soil structure and water retention capacity of high-altitude areas help regulate runoff and provide a stable supply of water resources [10]. The higher SOC content in high-altitude areas seen in this study is conducive to the long-term fixation of carbon, which has a potentially positive effect in mitigating climate change [72,73].

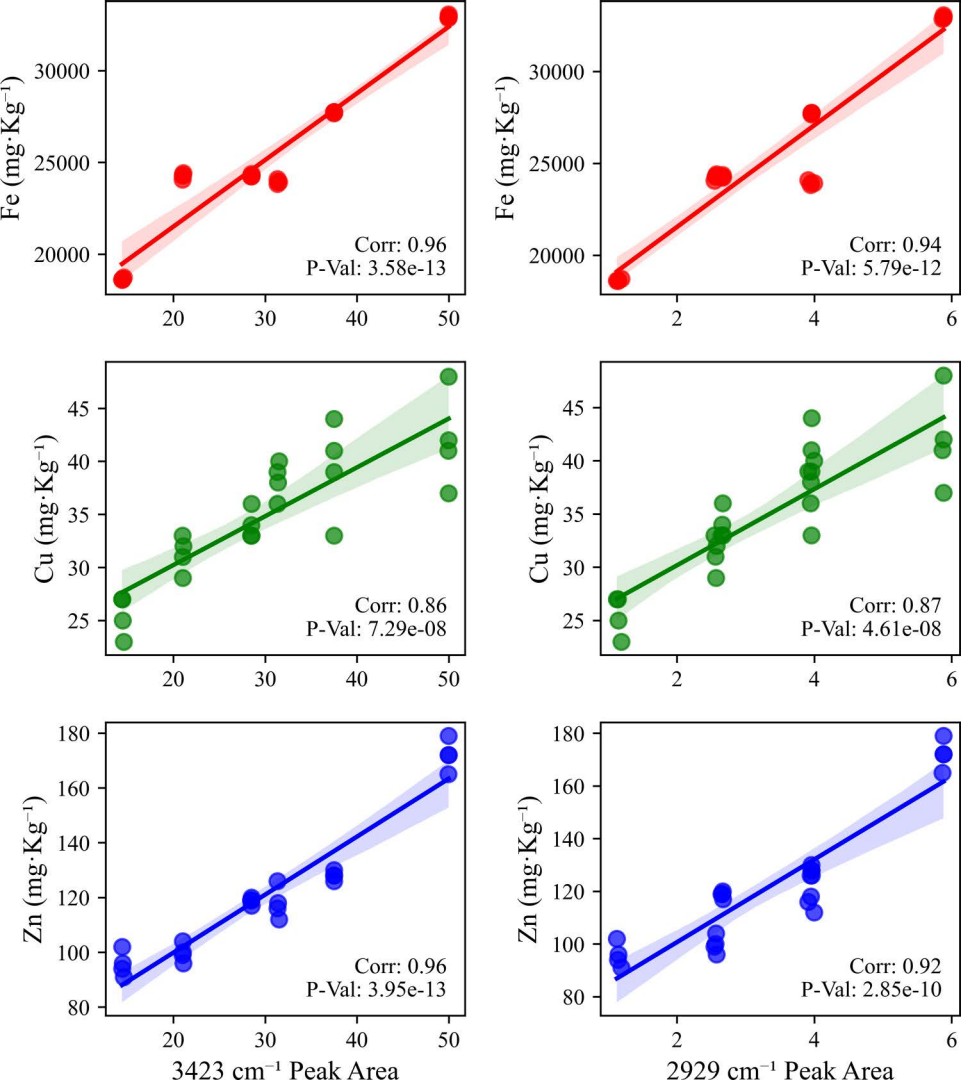

**Fig 11. Correlation Between Element Concentrations (Fe, Cu, Zn) and FTIR Peak Areas (3423 cm⁻¹ and 2929 cm⁻¹).**

### 4.2. Selection of sensitive factors and ecological significance of the Tongbai Mountain vertical zonation

This study used an innovative integrated analysis method that combined multiple statistical and machine learning techniques such as variance threshold feature selection, leave-one-out cross-validation, random forest feature importance, PCA, and correlation analysis. The advantage of this method is that it can comprehensively evaluate complex ecological data, overcome sample size limitations, identify robust indicators, and improve the reliability of results. By analysing the same dataset from multiple perspectives, this method can not only deal with linear and non-linear relationships, but also integrate multi-source data, providing a comprehensive perspective to understand the distribution of soil elements, minerals, and SOC in the mountain vertical zonation spectrum. Through a variety of feature selection and evaluation methods, this study provides a detailed analysis of the distribution of soil elements, minerals, and SOC in the vertical zonation spectrum of Mount Tongbai and their sensitive factors. Through variance threshold feature selection, 31 key features were selected from the original 33 features (Fig 10). Combined score cross-validation results showed that the model's mean

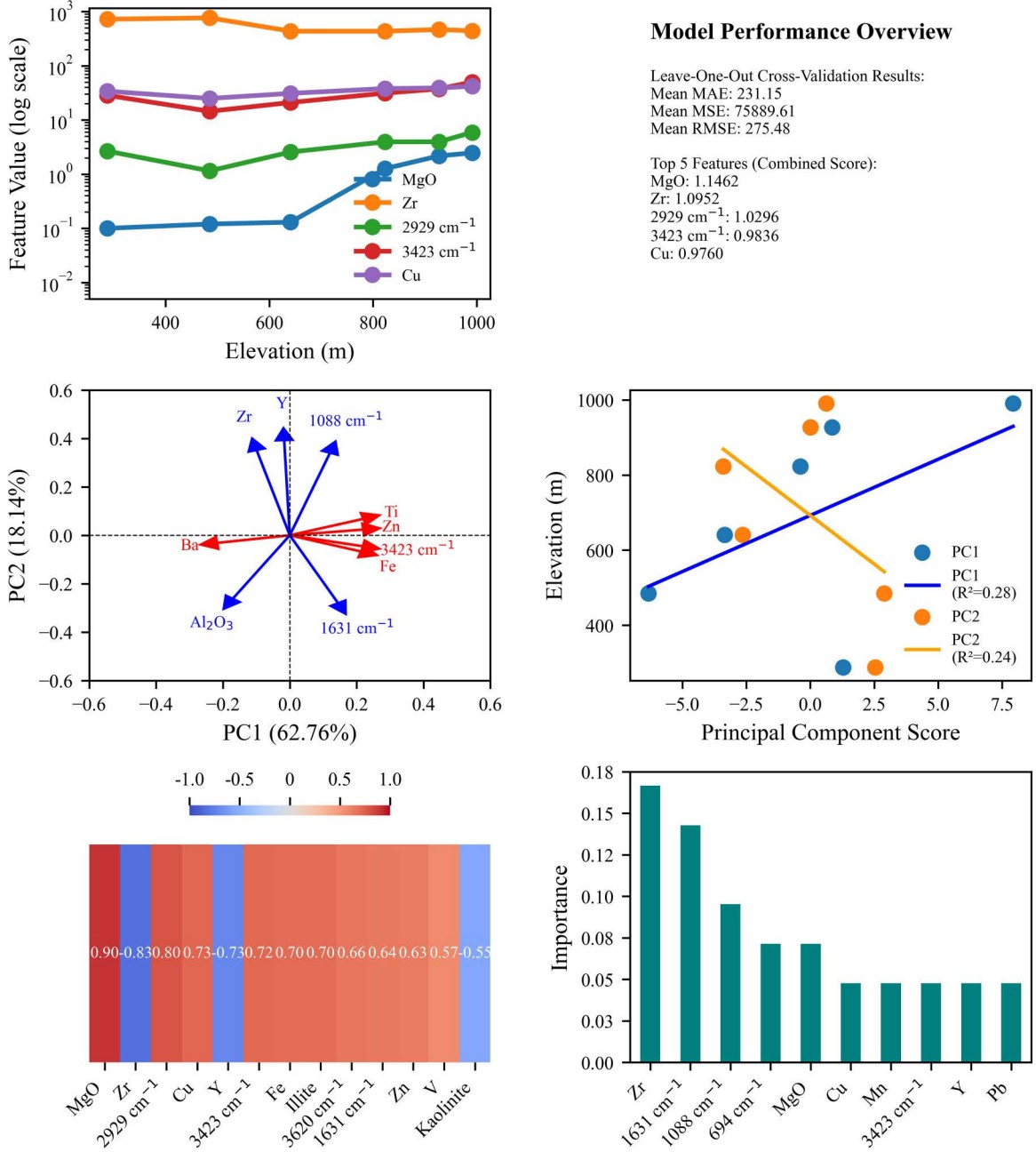

**Fig 12. Comprehensive analysis of elevation-dependent variations in elements, compounds, minerals, and FTIR spectral features.** (a) **Elevation trends of key features;** (b) **Model performance summary;** (c) **PCA loadings plot;** (d) Principal **components vs. elevation;** (e) **Correlation heatmap of significant features;** (f) Random **Forest feature importance ranking.**

absolute error (MAE) was 231.1517, the mean square error (MSE) was 75889.6139, and the root mean square error (RMSE) was 275.4807 (Fig 12). The comprehensive scoring method showed that the five most important features were MgO, Zr, 2929 cm⁻¹, 3423 cm⁻¹, and Cu (Fig 12a; Table 6), which played a key role in describing the variation of soil biogeochemical properties with altitude. The random forest feature importance analysis highlighted the importance of features such as Zr, 1631 cm⁻¹, and 1088 cm⁻¹, which may be closely related to soil mineral composition and SOC characteristics.

PCA results further revealed the importance of features such as Ti, Zn, 3423 cm⁻¹, and Ba in explaining the variability of the data (Fig 12c; Table 6). Of particular note, correlation analysis with altitude showed that MgO, Zr, 2929 cm⁻¹, Cu, and Y were the features that most correlated with altitude change (Fig 12e; Table 6). Among these, MgO and 2929 cm⁻¹ were strongly positively correlated with altitude, while Zr and Y were negatively correlated [2,5,12]. These findings provide important clues to understand changes in soil properties with altitude gradients, and comprehensively reflect the complexity of soil elements, minerals, and SOC distribution in the vertical belt spectrum of Tongbai Mountain and their relationship with altitude.

This study shows that SOC content and its characteristics changed significantly with altitude, and these changes had a significant impact on the migration and fixation behaviour of heavy metal elements, which in turn affected the environmental quality and ecological functions of mountain ecosystems. Firstly, the FTIR spectra of the high-altitude region showed enhanced peaks at 3423 cm⁻¹ (–OH or –NH groups) and 2929 cm⁻¹ (aliphatic C-H stretching vibrations) (Fig 8; Table 4). The higher content of aliphatic compounds in the high-altitude region may indicate that there are more heavy metal complexation sites in high-altitude soil. The higher SOC content at high altitudes was positively correlated with the enrichment trends of elements such as Cu, Zn, and Fe (Fig 10 and 11). This correlation may further confirm that SOC has a strong adsorption capacity for metal ions and can fix some heavy metal elements through complexation [74]. The immobilisation effect of organic-mineral complexes may reduce the bioavailability of heavy metal elements and thus reduce their potential harm to the ecosystem [75]. The higher SOC content in high-altitude areas may help delay the migration of heavy metal elements to downstream water bodies, thereby protecting the water quality of the water source area [76]. However, the organic-mineral complex-mediated heavy metal fixation may transform some areas (such as high-altitude SOC-enriched areas) into a 'sink' for heavy metals. With the impact of climate change and human activities, these fixed heavy metals may be released in the future, which in turn poses a potential threat to the ecosystem. SOC-mediated heavy metal fixation may effectively reduce the bioavailability and potential toxicity of heavy metals, thus protecting the stability of high-altitude ecosystems to a certain extent. Secondly, the altitude gradient of SOC types may lead to differences in the form of heavy metal elements in different altitude zones. The peak at 1631 cm⁻¹ (related to aromatic C=C or C=O vibration modes) was the largest in the middle altitude zone (Table 4), indicating that there may be more aromatic organic matter in this zone. Aromatic organic matter generally has a strong ability to bind metal ions, which may lead to differences in the types of heavy metals fixed in the middle-elevation region when compared to the high-elevation region. The regulatory effects of SOC on heavy metal behaviour are two-fold. Long-term monitoring and assessment of the dynamic changes in the interactions between SOC and heavy metals is crucial to predict and manage environmental risks in mountain ecosystems. Future research should further explore the migration mechanisms of different elements and the influence of soil microaggregates on element behaviour. This study reveals the complexity of soil biogeochemical processes in the vertical

**Table 6. Top 10 features ranked by combined score, random forest importance, PCA loadings, and correlation with elevation.**

| Rank | Combined Score | Random Forest Importance | PCA Loadings PC1 | PCA Loadings PC2 | Correlation with Elevation |
|------|----------------|--------------------------|-------------------|-------------------|----------------------------|
| 1 | MgO (1.1462) | Zr (0.1667) | Ti (0.2198) | Y (0.3930) | MgO (0.9034) |
| 2 | Zr (1.0952) | 1631 cm⁻¹ (0.1429) | Zn (0.2194) | Zr (0.3521) | Zr (−0.8292) |
| 3 | 2929 cm⁻¹ (1.0296) | 1088 cm⁻¹ (0.0952) | 3423 cm⁻¹ (0.2185) | 1088 cm⁻¹ (0.3406) | 2929 cm⁻¹ (0.7965) |
| 4 | 3423 cm⁻¹ (0.9836) | 694 cm⁻¹ (0.0714) | Ba (−0.2171) | 1631 cm⁻¹ (−0.2803) | Cu (0.7260) |
| 5 | Cu (0.9760) | MgO (0.0714) | Fe (0.2127) | $Al_2O_3$ (−0.2620) | Y (−0.7253) |
| 6 | 1631 cm⁻¹ (0.9311) | Cu (0.0476) | Illite (0.2122) | Pb (−0.2604) | 3423 cm⁻¹ (0.7175) |
| 7 | Fe (0.9128) | Mn (0.0476) | 2929 cm⁻¹ (0.2092) | 3620 cm⁻¹ (−0.2118) | Fe (0.7000) |
| 8 | Illite (0.9080) | 3423 cm⁻¹ (0.0476) | Microcline (−0.2056) | 800 cm⁻¹ (0.2109) | Illite (0.6958) |
| 9 | 3620 cm⁻¹ (0.8798) | Y (0.0476) | $K_2O$ (−0.2048) | V (−0.2009) | 3620 cm⁻¹ (0.6632) |
| 10 | Zn (0.8537) | Pb (0.0476) | Cu (0.2024) | 694 cm⁻¹ (−0.2004) | 1631 cm⁻¹ (0.6438) |

zonation of Mount Tongbai. By screening for sensitive factors in the vertical zonation and analysing the regulatory mechanisms of SOC on heavy metal behaviour, this study provides a scientific basis for environmental protection and resource management in mountain ecosystems. Of specific importance to climate change, this study provides an important theoretical basis for the protection of mountain ecological environments, prevention and control of soil pollution, and future climate change adaptation strategies.

Although the results of this study on Tongbai Mountain exhibit similarities with observations from other mountain ranges—suggesting that our conclusions may have broader applicability—caution is warranted when extrapolating these findings to different geographical contexts. It is important to note that the distributions of SOC and heavy metals may display distinct dynamic characteristics over varying timescales. For instance, SOC levels can respond rapidly (over periods of years to decades) to changes in vegetation or management practices, whereas heavy metal distributions tend to remain relatively stable, reflecting historical deposition patterns spanning hundreds to thousands of years. Consequently, future research should incorporate time-series data or radiocarbon dating techniques to further elucidate the turnover rates and stability of SOC and heavy metals, while also considering their long-term variations over extended periods.

## 5. Conclusions

This study systematically analysed soil biogeochemical processes in a typical mountain vertical belt spectrum in the East Asian north-south climatic transition zone. This revealed a collaborative evolution of soil mineral composition, SOC composition characteristics, and heavy metal element distribution, and thoroughly explored the driving mechanisms of the altitude gradient on biogeochemical processes. This study combined the vertical belt spectrum with biogeochemical processes to systematically reveal the driving mechanisms of the altitude gradient on the collaborative evolution of soil minerals, SOC, and heavy metal elements. The main conclusions are as follows: (1) The composition of soil minerals showed a clear gradient with altitude. The content of Quartz was higher at low and high altitudes, showing a 'U'-shaped distribution; the content of feldspar minerals (microcline, albite) was highest in the middle altitude area, and then significantly decreased in the high-altitude area. Illite increased with altitude. This reflects the combined effects of the degree of weathering, the source of the material, and the climatic conditions on the distribution of minerals. (2) Altitude change was an important factor affecting SOC and its stabilization processes. The content of SOC increased with increasing altitude. The content of hydroxyl, amino, and aliphatic compounds in SOC at high altitude areas increased significantly, while aromatic SOC were enriched in middle altitude areas. FTIR analysis showed that the type of functional group in SOC changed significantly with altitude, reflecting the combined effects of climatic conditions, vegetation type, and microbial activity. (3) The content of heavy metal elements (such as Fe, Ti, Cu, Zn) in soil increased with increasing altitude. The low temperature, high humidity, and higher SOC content at high altitudes enhanced the adsorption and fixation capacity of the soil for heavy metal elements, which affected the migration and accumulation of heavy metals. (4) SEM analysis showed that the structure of microaggregates in high-altitude soil was more stable, the particles were closely bonded, and the pore structure was complex. This may be related to the formation of organic-mineral complexes between SOC and clay minerals, which enhances the stability of soil aggregates. (5) By comprehensively using methods such as correlation analysis, random forest feature importance, and PCA, MgO, Zr, $2929\,cm^{-1}$, $3423\,cm^{-1}$, and Cu were selected as sensitive biogeochemical factors in the vertical band spectrum. Changes in these factors reflect the significant influence of the altitude gradient on soil properties, which deepens the understanding of biogeochemical processes in mountain ecosystems. In the context of climate change, this study provides important theoretical findings and practical guidance for ecological and environmental protection, soil pollution control, and the formulation of future climate change adaptation strategies.

## Author contributions

**Conceptualization:** Chunjie Li.

**Data curation:** Chunjie Li, Songhao Shang.

Formal analysis: Chunjie Li.

Investigation: Chunjie Li, Shili Guo.

Methodology: Chunjie Li.

Software: Chunjie Li.

Validation: Songhao Shang.

Writing – original draft: Chunjie Li, Shili Guo.

Writing – review & editing: Chunjie Li, Shili Guo, Songhao Shang.

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
