## [Decision Letter · Decision Letter 0]

10 Mar 2025

Dear Dr. Li,

Thank you for submitting your manuscript to PLOS ONE. After careful consideration, we feel that it has merit but does not fully meet PLOS ONE’s publication criteria as it currently stands. Therefore, we invite you to submit a revised version of the manuscript that addresses the points raised during the review process.

We look forward to receiving your revised manuscript.

Kind regards,

Erika Kothe

Academic Editor

PLOS ONE

Journal Requirements:

National Nature Science Foundation of China [grant number 41601614].

5. We note that your Data Availability Statement is currently as follows: All relevant data are included within the manuscript and its Supporting Information files.

6. We note that Figure 1 in your submission contain copyrighted images. All PLOS content is published under the Creative Commons Attribution License (CC BY 4.0), which means that the manuscript, images, and Supporting Information files will be freely available online, and any third party is permitted to access, download, copy, distribute, and use these materials in any way, even commercially, with proper attribution. For more information, see our copyright guidelines: http://journals.plos.org/plosone/s/licenses-and-copyright.

7. We note that Figure 1 in your submission contain [map/satellite] images which may be copyrighted. All PLOS content is published under the Creative Commons Attribution License (CC BY 4.0), which means that the manuscript, images, and Supporting Information files will be freely available online, and any third party is permitted to access, download, copy, distribute, and use these materials in any way, even commercially, with proper attribution. For these reasons, we cannot publish previously copyrighted maps or satellite images created using proprietary data, such as Google software (Google Maps, Street View, and Earth). For more information, see our copyright guidelines: http://journals.plos.org/plosone/s/licenses-and-copyright.

b . If you are unable to obtain permission from the original copyright holder to publish these figures under the CC BY 4.0 license or if the copyright holder’s requirements are incompatible with the CC BY 4.0 license, please either i) remove the figure or ii) supply a replacement figure that complies with the CC BY 4.0 license. Please check copyright information on all replacement figures and update the figure caption with source information. If applicable, please specify in the figure caption text when a figure is similar but not identical to the original image and is therefore for illustrative purposes only.

Additional Editor Comments :

The reviewers gave helpful comments. Especially the remark about generalization being very limited if only one time sampling was used is a severe shortcoming for a journal that hopes to attract readers interested in general microbiology. Only if the data can be compared to other publications, or preferably, a second time point, the manuscript can become acceptable for PlosOne.

Reviewers' comments:

Reviewer's Responses to Questions

**Comments to the Author**

1. Is the manuscript technically sound, and do the data support the conclusions?

Reviewer #1: Yes

Reviewer #2: Yes

2. Has the statistical analysis been performed appropriately and rigorously?

Reviewer #1: Yes

Reviewer #2: I Don't Know

3. Have the authors made all data underlying the findings in their manuscript fully available?

Reviewer #1: Yes

Reviewer #2: Yes

4. Is the manuscript presented in an intelligible fashion and written in standard English?

Reviewer #1: Yes

Reviewer #2: Yes

Reviewer #1: The study used a variety of advanced analytical techniques to comprehensively characterize the mineral composition, soil organic carbon (SOC), and heavy metal element distribution of soils at different altitudes in Mount Tongbai. The manuscript proposes a comprehensive scoring method to evaluate the sensitive factors of soil biogeochemical characteristics. This method has some innovation and practicality in ecological data analysis. The data is detailed, and the methods are scientific. It is believed that reliable research results can be provided. Although the research scope is limited to this specific region, it may limit the generalizability and generalizability of the research results. Moreover, based solely on one-time sampling and analysis, lacking long-term monitoring data, it may not fully reflect the dynamic changes in soil biogeochemical processes. I suggest the authors check the details of the manuscript carefully. For example, what does the “SThe structure” in Line 230 mean? In addition, the discussion part should be improved, and some similar previous studies should be cited and discussed.

Reviewer #2: Dear Authors

I have several comments regarding the X-ray fluorescence analysis.

1. If you used reference materials to construct calibration curves for determining elements, you should list them, including manufacturers and analyte ranges. If the calculation was carried out using a standardless approach, it should also be indicated how exactly this was done. However, in this case, an analysis of the reference material is also necessary to estimate the accuracy of applied technique (reference materials are quite available for soil composition).

2. Table 3 is unclear to me. What does 0.00±1.12 mean? Why is the confidence interval for Mg so large, while for Al these values are several times smaller? Was Na which is also a major element for soils, analyzed?

3. Before Table 5, the LOD values for each element and how they were calculated must be indicated. Was fluorine analyzed? Some values, for example, for barium 435±399, also raise questions. What conclusions can be made if the standard deviation is almost 100 percent? Are you sure that barium is present in these samples? Why is iron not classified as a major element and the accuracy of its determination (according to the table) is significantly higher than, for example, for calcium?

I recommend describing in more detail the X-ray fluorescence method, as well as critically approaching obtained results.

**Do you want your identity to be public for this peer review?** For information about this choice, including consent withdrawal, please see our Privacy Policy

Reviewer #1: No

Reviewer #2: No

---

## [Author Response · Author response to Decision Letter 1]

31 Mar 2025

Response to Editor Comments

Dear Editor,

Thank you for providing us with valuable feedback and suggestions, which have significantly improved the quality of our manuscript. Here we submit a revised manuscript with the title “Synergistic Evolution of Soil Microaggregates Biogeochemical Processes Driven by Elevation Gradients in Tongbai Mountain”, which has been modified according to the suggestions carefully. The followings are point-to-point responses to the comments.

Comment 1: Please ensure that your manuscript meets PLOS ONE's style requirements, including those for file naming. The PLOS ONE style templates can be found at https://journals.plos.org/plosone/s/file?id=wjVg/PLOSOne_formatting_sample_main_body.pdf and https://journals.plos.org/plosone/s/file?id=ba62/PLOSOne_formatting_sample_title_authors_affiliations.pdf

Response 1: We have revised the manuscript according to the PLOS ONE formatting style, including the main text structure and file naming requirements. All revised files conform to the templates provided in the formatting guidelines.

Comment 2: In your Methods section, please provide additional information regarding the permits you obtained for the work. Please ensure you have included the full name of the authority that approved the field site access and, if no permits were required, a brief statement explaining why.

Response 2: We have added a statement in the Methods section indicating that field sampling was carried out with the approval of the Tongbai Mountain National Nature Reserve Administration Bureau. (line 242-243). No special permits were required as the study did not involve protected species or private land.

Comment 3: Thank you for stating the following financial disclosure:

National Nature Science Foundation of China [grant number 41601614].

Response 3: We have included the following statement in the manuscript and cover letter: "The funders had no role in study design, data collection and analysis, decision to publish, or preparation of the manuscript."

Comment 4a & 5: Clarify data access restrictions or upload data to a repository. Confirm data availability.

Response4a & 5: There are no ethical or legal restrictions on sharing the data. All relevant raw data, including values behind graphs and statistical measures, have now been uploaded as Supporting Information files and to an open repository [repository link/DOI to be inserted upon upload]. The Data Availability Statement has been updated accordingly. All relevant data are contained within the article.

Comment 6 & 7: Figure 1 includes copyrighted [map/satellite] images. You must obtain permission or replace them.

Response 6 & 7: We have removed the previously used copyrighted image in Figure 1 and replaced it with a schematic diagram titled “Schematic diagram of vertical vegetation zonation along an altitudinal gradient in Tongbai Mountain.” The updated figure complies with the CC BY 4.0 license, and the figure caption has been revised to include appropriate attribution.

Sincerely,

Chunjie LI

Response to Reviewer 1 Comments

Dear Reviewer,

Thank you for providing us with valuable feedback and suggestions, which have significantly improved the quality of our manuscript. Here we submit a revised manuscript with the title “Synergistic Evolution of Soil Microaggregates Biogeochemical Processes Driven by Elevation Gradients in Tongbai Mountain”, which has been modified according to the suggestions carefully. The followings are point-to-point responses to the comments.

Comment 1: The study used a variety of advanced analytical techniques to comprehensively characterize the mineral composition, soil organic carbon (SOC), and heavy metal element distribution of soils at different altitudes in Mount Tongbai. The manuscript proposes a comprehensive scoring method to evaluate the sensitive factors of soil biogeochemical characteristics. This method has some innovation and practicality in ecological data analysis. The data is detailed, and the methods are scientific. It is believed that reliable research results can be provided. Although the research scope is limited to this specific region, it may limit the generalizability and generalizability of the research results..

Response 1: Thank you for recognizing the innovation and scientific merit of our methodology. We acknowledge that focusing solely on Tongbai Mountain may limit the generalizability of our results. To address this, we have significantly expanded the Discussion section by comparing our findings with studies from various mountain ranges—including the Tatra Mountains, the Andes, the Helan Mountains, the Lushan Mountains, the Dabie Mountains, the Bamtou Mountains, the Karakorum Mountains, Mount Everest, the Alps, Luoji Mountain, and the Hengduan Mountains—and by citing multiple studies (e.g., Cao et al., 2024; Du et al., 2014; Li et al., 2018; Magnani et al., 2018; Ouyang et al., 2021; Shedayi et al., 2016; Sun et al., 2023; Tsozué et al., 2019; Wu et al., 2023; Zechmeister, 1995) to provide a broader ecological and geochemical context for our results. (line 525-532; line 543-557; line 566-587; line 611-615). Through these efforts, we aim to enhance the broader applicability of our research outcomes and provide a replicable analytical framework and theoretical support for future studies in similar settings.

Comment 2: Based solely on one-time sampling and analysis, lacking long-term monitoring data, it may not fully reflect the dynamic changes in soil biogeochemical processes.

Response 2: Thank you for your important comment regarding the one-time sampling and the absence of long-term monitoring data. We acknowledge that our study is based on a single sampling event, and we have added a statement in the Discussion section to underscore that long-term monitoring is essential for capturing the temporal variability in soil biogeochemical processes. In future work, we plan to conduct multi-year sampling campaigns to address this issue more effectively. Although our results from Tongbai Mountain show similarities with observations from other mountain ranges—suggesting that our conclusions may have broader applicability—we caution against extrapolating these findings to different geographical contexts without further investigation. It is also important to note that the distributions of soil organic carbon (SOC) and heavy metals may exhibit distinct dynamic characteristics over various timescales. For example, SOC levels can respond rapidly (within years to decades) to changes in vegetation or management practices, whereas heavy metal distributions tend to remain relatively stable, reflecting historical deposition patterns that span hundreds to thousands of years. (line 711-722).Therefore, incorporating time-series data or radiocarbon dating techniques in future research will be essential for further elucidating the turnover rates and long-term stability of SOC and heavy metals.

Comment 3: I suggest the authors check the details of the manuscript carefully. For example, what does the “SThe structure” in Line 230 mean?

Response 3: We thank the reviewer for spotting this typographical error. The text "SThe structure" is a typo and has been corrected to "The structure" in the revised manuscript (Section 2.3, SEM Analysis). (line 274).

Comment 4: The discussion part should be improved, and some similar previous studies should be cited and discussed.

Response 4: We agree with the reviewer and have expanded the Discussion section accordingly. In the revised version, we compare our findings with studies from other mountainous regions, including the Tatra Mountains and the Andes. We now reference multiple studies (e.g., Cao et al., 2024; Du et al., 2014; Li et al., 2018; Magnani et al., 2018; Ouyang et al., 2021; Shedayi et al., 2016; Sun et al., 2023; Tsozué et al., 2019; Wu et al., 2023; Zechmeister, 1995) to provide a broader ecological and geochemical context for our results. (line 525-532; line 543-557; line 566-587; line 611-615). These additions help to highlight both the uniqueness and the generality of the elevation-driven patterns observed in our study.

Sincerely,

Chunjie LI

Response to Reviewer 2 Comments

Dear Reviewer,

Thank you for providing us with valuable feedback and suggestions, which have significantly improved the quality of our manuscript. Here we submit a revised manuscript with the title “Synergistic Evolution of Soil Microaggregates Biogeochemical Processes Driven by Elevation Gradients in Tongbai Mountain”, which has been modified according to the suggestions carefully. The followings are point-to-point responses to the comments.

Comment 1: If you used reference materials to construct calibration curves for determining elements, you should list them, including manufacturers and analyte ranges. If the calculation was carried out using a standardless approach, it should also be indicated how exactly this was done. However, in this case, an analysis of the reference material is also necessary to estimate the accuracy of applied technique (reference materials are quite available for soil composition).

Response 1: We appreciate the reviewer’s detailed and professional feedback. In the revised manuscript (Section 2.5), we have clarified that XRF measurements were conducted using a PANalytical AXIOS WDXRF spectrometer. This instrument is equipped with SuperQ 4.0 quantitative analysis software, which automatically handles peak intensity calculations, Compton scatter internal standard correction, spectral interference correction, and matrix correction before displaying the analysis results. Calibration was performed using certified reference materials (GBW07401 and GBW07404 from the Chinese National Institute of Metrology) that encompass the major and trace element concentration ranges relevant to soil. The analytical accuracy was verified by repeatedly testing these reference materials and comparing the results with their certified values, yielding recovery rates generally between 90% and 110%. This additional information has been included to enhance transparency and ensure data quality. (line289-303).

Comment 2: Table 3 is unclear to me. What does 0.00±1.12 mean? Why is the confidence interval for Mg so large, while for Al these values are several times smaller? Was Na which is also a major element for soils, analyzed?

Response 2: We appreciate the reviewer’s insightful comments regarding the presentation of our results. MgO and similar compounds were also measured using XRF. Compared with high atomic weight elements, XRF exhibits lower sensitivity for low atomic weight elements due to their low characteristic X-ray energies, weak fluorescence yields, and the ease with which their signals are absorbed by air and detector windows. As a result, elements such as Na, Mg, and Al have higher detection limits and greater measurement uncertainties. The values, such as 0.00 ± 1.12 reported in Table 3, indicate that the measured concentrations of MgO in some samples were close to or below the instrument’s detection limit, leading to high uncertainty. The large standard deviation reflects the very low MgO concentrations in those specific samples (near 0%). In contrast, Al₂O₃ levels were consistently higher and fell within the instrument’s optimal detection range, resulting in lower variability. Regarding sodium (Na), due to limitations in our XRF setup—specifically, its low fluorescence yield and high volatility—Na was not analyzed. This explanation has been added to the Methods section. (line304-316).

Comment 3: Before Table 5, the LOD values for each element and how they were calculated must be indicated. Was fluorine analyzed? Some values, for example, for barium 435±399, also raise questions. What conclusions can be made if the standard deviation is almost 100 percent? Are you sure that barium is present in these samples? Why is iron not classified as a major element and the accuracy of its determination (according to the table) is significantly higher than, for example, for calcium?

Response 3: We thank the reviewer for the insightful comments. By repeatedly measuring the reference materials and comparing the results with their certified values, we achieved recovery rates generally between 90% and 110%, indicating that the method has high analytical precision, although its accuracy for trace elements is limited. The limit of detection (LOD) for each element was determined using the 3σ criterion (three times the standard deviation of blank measurements). The high standard deviation observed for barium (e.g., 435 ± 399) reflects both significant spatial variability in its concentration and the fact that some samples were near the detection limit; we have acknowledged this uncertainty in the revised discussion. Furthermore, the XRF results for trace elements should be considered qualitative. Although iron (Fe) is typically classified as a heavy metal, it also functions as a major oxide-forming element in soil. We have now clarified in both the Methods and Discussion sections that, despite Fe not being grouped under "major elements," its role and high analytical accuracy (due to its strong fluorescence yield and abundant concentration) render it comparable to the major oxides. (line304-316).

Sincerely,

Chunjie LI

---

## [Decision Letter · Decision Letter 1]

15 Apr 2025

Dear Dr. Li,

Thank you for submitting your manuscript to PLOS ONE. After careful consideration, we feel that it has merit but does not fully meet PLOS ONE’s publication criteria as it currently stands. Therefore, we invite you to submit a revised version of the manuscript that addresses the points raised during the review process.

We look forward to receiving your revised manuscript.

Kind regards,

Erika Kothe

Academic Editor

PLOS ONE

Additional Editor Comments:

The answers to the reviewers query did not completely meet the expectations. Please be very careful to commit to diligent following standard procedures.

Reviewers' comments:

Reviewer's Responses to Questions

**Comments to the Author**

Reviewer #2: All comments have been addressed

2. Is the manuscript technically sound, and do the data support the conclusions?

Reviewer #2: Yes

3. Has the statistical analysis been performed appropriately and rigorously?

Reviewer #2: I Don't Know

4. Have the authors made all data underlying the findings in their manuscript fully available?

Reviewer #2: Yes

5. Is the manuscript presented in an intelligible fashion and written in standard English?

Reviewer #2: Yes

Reviewer #2: Dear Authors

I still have some comments.

Presenting the results as 0.00±1.12 is completely unacceptable. If this content (or the content of any other element being determined) was close to the detection limit, you should write <lod, better="" or="" yet="">

Despite the adequacy of your answer, I recommend adding information about the analysis of certified reference materials to the text. The part of your answer is quite acceptable for it.</lod,>

**Do you want your identity to be public for this peer review?** For information about this choice, including consent withdrawal, please see our Privacy Policy

Reviewer #2: No

---

## [Author Response · Author response to Decision Letter 2]

16 Apr 2025

Dear Reviewer,

Thank you for your insightful feedback and valuable suggestions, which have greatly contributed to improving the quality of our manuscript. We are pleased to submit a revised version of our manuscript titled “Synergistic Evolution of Soil Microaggregates Biogeochemical Processes Driven by Elevation Gradients in Tongbai Mountain”. The manuscript has been carefully revised in accordance with your comments. Below are our point-by-point responses to your suggestions.

Comment 1: Presenting the results as 0.00±1.12 is completely unacceptable. If this content (or the content of any other element being determined) was close to the detection limit, you should write.

Response 1: Thank you for your valuable comment. We acknowledge that presenting results as "0.00±1.12" without specifying the detection limit (LOD) can be misleading. In our revised manuscript, we have now clearly indicated that these values are below the detection limit (LOD), and we have revised the results accordingly to reflect this. (line309-315). The relevant results are now presented as “< LOD” (below the limit of detection) to better align with scientific reporting standards for data close to the detection limit. We have made these changes in both the manuscript text and Table 3 to avoid any potential misinterpretation. We appreciate your suggestion, which helps to improve the accuracy of our presentation.

Comment 2: Despite the adequacy of your answer, I recommend adding information about the analysis of certified reference materials to the text. The part of your answer is quite acceptable for it.

Response 2: We have included details in the manuscript on the calibration of our methods using certified reference materials (GBW07401 and GBW07404 from the Chinese National Institute of Metrology), which cover major and trace element concentrations, as well as major element oxide concentrations relevant to soil analysis. The analytical accuracy was validated through repeated testing, with recovery rates typically ranging from 90% to 110%.

This information has been added to the Methodology section (lines 303-308) of the manuscript for transparency and clarity. The specific text added is as follows:

“Calibration was carried out using certified reference materials (GBW07401 and GBW07404 from the Chinese National Institute of Metrology), which cover the major and trace element concentration ranges relevant to soil. The analytical accuracy was validated through repeated testing of these reference materials, with the results compared to their certified values, yielding recovery rates typically ranging from 90% to 110%.”

We believe that this addition strengthens the rigor of our study and addresses your concern regarding the quality assurance of our analytical methods.

We would like to thank you for your invaluable feedback, which has helped improve the manuscript. We believe that the revisions made in response to your suggestions have significantly enhanced the quality of our work. We look forward to your further feedback.

Sincerely,

Chunjie LI

---

## [Editor Report · Decision Letter 2]

13 May 2025

Synergistic Evolution of Soil Microaggregates Biogeochemical Processes Driven by Elevation Gradients in Tongbai Mountain

PONE-D-25-05130R2

Dear Dr. Li,

We’re pleased to inform you that your manuscript has been judged scientifically suitable for publication and will be formally accepted for publication once it meets all outstanding technical requirements.

Kind regards,

Erika Kothe

Academic Editor

PLOS ONE

---

## [Editor Report · Acceptance letter]

PONE-D-25-05130R2

PLOS ONE

Dear Dr. Li,

I'm pleased to inform you that your manuscript has been deemed suitable for publication in PLOS ONE. Congratulations! Your manuscript is now being handed over to our production team.

Kind regards,

on behalf of

Prof. Dr. Erika Kothe

Academic Editor

PLOS ONE